# Digital Finance and Corporate Sustainability Performance: Promoting or Restricting? Evidence from China's Listed Companies

**Sumin Hu** 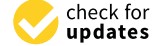**, Qi Zhu, Xia Zhao * and Ziyue Xu**

Business School, Suzhou University of Science and Technology, Suzhou 215009, China; alvahsm@usts.edu.cn (S.H.); 2211051022@post.usts.edu.cn (Q.Z.); 20200417126@post.usts.edu.cn (Z.X.)
* Correspondence: zhaoxia@usts.edu.cn; Tel.: +86-1367-949-5519

**Abstract:** The development of internet platforms and information technology has accelerated the transformation of conventional finance. Emerging digital finance is expected to optimize the allocation of credit resources and thereby promote a sustainable transition for corporations. However, whether, and to what extent, digital finance empirically affects this process is still not well understood. This paper investigates the role of digital finance in promoting corporate sustainability performance by exploring its impact on green enterprise innovation and its mechanism using a two-way fixed effects model and a mediating effects model. The findings suggest the following: (i) The impact of digital finance on the sustainable performance of enterprises follows a U-shaped (coef. = 0.00, t = 2.43) pattern, where digital finance initially restricts and then promotes the sustainable performance of enterprises. This conclusion remains robust even after considering endogeneity. (ii) The mechanism analysis indicates that digital finance enhances sustainability performance by reducing corporate financial volatility (coef. = −0.00, t = −4.06) and promoting long-term performance growth (coef. = 6.69, t = 4.88). (iii) The positive effects of digital finance on sustainability performance are more significant for non-state-owned firms (coef. = 0.00, t = 5.42), firms located in cities with a lower GDP per capita (coef. = 0.00, t = 2.40), and smaller firms (coef. = −0.00, t = −2.59) in their initial stages. These results imply that China should accelerate digitization in the financial markets and thus further develop its potential for sustainable development.

**Keywords:** digital finance; sustainability performance; corporate resilience; mediating effect

## 1. Introduction

In recent years, mediation of the synergy between ecological optimization and economic development has become a global challenge, and it has had a significant impact on enterprises' efforts to achieve sustained and competitive business advantages [1,2]. Pro-environmental behaviors, social responsibility awareness, and governance capacity are recognized as key factors strongly associated with enterprises' sustainability performance [3–5]. This suggests that long-term and adequate funding may be a prerequisite for firms to meet the demands of sustainable development [2,6], while financial constraints often hinder progress toward plural sustainability [7,8]. Therefore, efficient and sustainable financial support is essential for firms to achieve their sustainability goals [9,10]. However, friction between traditional financial institutions and firms, caused by information asymmetry and agency problems, can significantly increase financing costs [11,12]. Therefore, an effective approach to overcoming financial constraints is necessary for firms to achieve sustainable development.

Information technologies such as big data, cloud computing, blockchain technologies, artificial intelligence, and other digital technologies have transformed traditional finance services [13,14]. These technologies have enabled firms to have greater access to finance [2,15]. Theoretically, digital finance is expected to accelerate lending procedures

by efficiently and accurately processing loan and mortgage applications [16,17], which can help firms to achieve their green and sustainable development targets. Digital finance can significantly reduce information asymmetry and agency problems compared to traditional finance through the use of information and data sources from third-party assessments [17,18]. Additionally, digital finance has the potential to decrease the investment and financing risks of firms in the sustainable development process [2,6]. However, digital finance may also have adverse effects on the environment through individuals' strategic behaviors, such as a substantial increase in energy demand [19,20]. Moreover, the specific impact of digital finance on corporate sustainability performance remains controversial and varies greatly across different regions, as the digitalization of financial institutions is still in its early stages in China [21,22]. Therefore, it is of great practical importance to explore whether and to what extent digital finance affects corporate sustainability performance.

This paper aims to investigate the impact of digital finance on corporate sustainability performance, with a particular focus on the role of corporate resilience. While previous studies have mainly examined the influences of financing capacity and green innovation [23,24], recent studies on corporate resilience provide a new perspective for understanding the mechanism of digital finance in regard to sustainable performance [18,25], particularly in light of COVID-19 and China's commitments to cap and reduce carbon emissions. Available funds can enhance corporate resilience by providing essential resources with which to reshape firms' production and operations [18,26,27]. Moreover, enterprises with greater corporate resilience are more likely to exhibit lower financial volatility and higher growth performance [28] and the ability to effectively integrate and allocate crucial resources towards sustainable development. Therefore, we expect that corporate resilience will mediate the relationship between digital finance and corporate sustainability performance.

This study can contribute to the present literature in several ways: First, previous studies have mainly focused on analyzing the influencing factors of corporate sustainable development in the contexts of environmental regulation, government subsidies, and technical advancement [29,30]. As an emerging financial service model, digital finance significantly affects firms' diversified goals towards sustainable transition [31]. Regional digital finance, a novel perspective, is therefore applied in this paper to promote corporate sustainability performance. Second, little is known about whether and to what extent digital finance affects corporate sustainability performance. This study enriches the previous literature by providing empirical evidence for the nonlinear impact of digital finance on sustainability performance. Meanwhile, the mediating role of corporate resilience offers a new angle, revealing the "black box" of the theoretical relationship between digital finance and corporate sustainable development. Third, several methods are applied to address endogeneity and the robustness of the baseline model.

This paper is organized as follows: Section 2 analyzes the literature on the relationship between digital finance and corporate sustainability performance, as well as the literature on the mediating effect of corporate resilience. In Section 3, we describe the empirical strategy employed in this study. Section 4 presents the empirical findings and corresponding discussion. Finally, Section 5 outlines the conclusions, research limitations, and future directions of study.

## 2. Research Review and Hypothesis

### 2.1. Digital Finance and Corporate Sustainability Performance

The concept of digital finance has gained significant attention due to the deep integration of digital technology into the financial industry. Digital finance is distinct from traditional finance, as it allows businesses to access financial services through digital channels efficiently [18,32]. With the help of big data and algorithms, digital finance makes traditional payment, investment, and lending services more efficient, accessible, affordable, and commercially sustainable [2,33]. To be specific, the use of new information technologies in data collection and analysis facilitates loan procedures and provides efficient, accurate, and personalized customer service [18,34]. Previous studies on digital finance focused on

its impacts on economic growth [35,36] and financing capacity [37]. Recently, emerging research has explored the impacts of digital finance on issues related to carbon emissions, green innovation, inclusive growth, and alternative sustainability [2,18,27,38] (See Table 1).

**Table 1.** Overview of contributions to this topic.

| Author(s) | Assumption | Method | Empirical Result |
|---|---|---|---|
| Broby et al., (2018) [35] | The joint collaboration in fintech will benefit output and labor productivity. | Case analysis; theoretic deduction. | New fintech methods and innovations positively increase shareholder value, productivity, and employment. |
| Li and Liu, (2021) [14,36] | Digital-inclusive finance can indirectly promote the development of a green economy by increasing the concentration of producer services and optimizing the upgrading of the industrial structure. | Two-way fixed effects model; threshold effect mode; instrumental variable model (IV); SYS-GMM. | There is a significant positive U-shaped nonlinear relationship between digital-inclusive finance and green development. |
| Liu et al., (2022) [39] | Digital financial inclusion is able to promote China's economic growth through promoting small- and medium-sized enterprise entrepreneurship and stimulating residents' consumption. | VAR model; threshold regression model; multiple intermediary models. | The impact of digital financial inclusion development on economic growth has a significant internet threshold effect. |
| Cao et al., (2021) [40] | Green technology innovation is the transmission path through which digital finance affects energy–environmental performance. | Two-way fixed effects model; mediating effects model; difference in difference (DID) model. | Digital finance significantly improves China's energy–environmental performance, and green-tech innovation plays an important intermediary role in this process. |
| Wang et al., (2022). [41] | Digital financial inclusion may affect carbon emissions between cities through its breadth of coverage, depth of use, and level of digitization. | Spatial econometric model; mediating effects model. | Digital financial inclusion positively impacts CO2 emissions of local cities but negatively impacts neighboring cities. |
| Lin and Ma, (2022) [2] | Digital finance can significantly enhance the efficiency of capital allocation and reduce financing costs, which is beneficial for alleviating the financing constraints of green innovation. | Two-way fixed effects model; mediating effects model | Digital finance can improve the quantity and quality of green technological innovation; digital finance indirectly improves green innovation mainly by alleviating financing constraints. |
| Zhang and Liu, (2022) [1] | Digital finance can help to solve thefunding dilemma of green-tech innovation and can be combined well with carbon emission efficiency for better carbon emission efficiency. | Two-way fixed effects model; spatial econometric model. | The synergistic effect of digital finance and green technological innovation plays a significant role in promoting local carbon emission efficiency but suppresses carbon emission efficiency in surrounding cities to some extent. |

With the multi-objectives and complex risks involved in sustainable development, enterprises require long-term and stable funding to achieve good environmental, social, and governance performance [42] and to improve green technology and low-carbon equipment [8,9]. However, traditional financial services have a limited impact in alleviating financial constraints due to their asset-based mortgage loan model and complex approval procedures, making it challenging for micro-, small-, and medium-sized enterprises to obtain financial support. Digital finance can help businesses to deal with these financial constraints by improving credit allocation efficiency, offering liquidity support, and facilitating efficient, accurate, and individualized customer service [16,43]. Accordingly, digital finance broadens firms' access to external resources, helping them to achieve sustainable development goals while enhancing their overall financial stability and competitive advantage.

Furthermore, the research conducted by Liu et al. (2021) [36] and Li et al. (2021) [14] suggests that there exists a positive U-shaped non-linear correlation between digital finance and sustainable development in China. Specifically, the impact of digital finance has not yet

been fully realized due to its global infancy. This indicates that the process of digitalization in financial institutions may require a considerable amount of public resources, which could potentially be detrimental to sustainable corporate development. However, as digital finance advances, the marginal cost of financing decreases, making it easier for firms to access stable and long-term financial support [14,15,23]. This suggests that digital finance has the potential to accelerate the transition toward corporate sustainability when it reaches a certain level of development. Moreover, the level of digitalization in financial services can vary significantly across different regions due to economic imbalances [21,22]. As such, the positive effects of digital finance on sustained technological progress, organizational change, and environmental sustainability need further investigation. Consequently, we propose Hypothesis 1:

**H1.** *There is a U-shaped relationship between digital finance and corporate sustainability performance.*

*2.2. The Mediating Role of Corporate Resilience*

According to previous studies, sustained financial support is expected to strengthen the corporate resilience of firms, allowing them to actively respond to systemic crises [18,27] and thus achieve sustainable growth. Corporate resilience is often defined as a firm's ability to actively respond to external and internal shocks [44] and is considered as an essential capability for achieving sustainability [18,45]. Firms with a higher level of resilience tend to experience lower financial volatility and higher sales growth [28], ensuring their safe, stable, and long-term operation and production. A stable financial status provides long-term financial support for enterprises, enabling them to meet the diverse requirements of stakeholders regarding pro-environmental behaviors, social responsibility, and governance capacity. Sustainable sales growth is often associated with higher market sensitivity [28], which enables firms to better attract and retain crucial resources needed for sustainability, such as gaining more credit resources. Furthermore, corporate resilience, specifically defined as sustainable sales growth and financial stability, pushes companies to improve their operational and learning abilities [39], which enables enterprises to more accurately acquire and use these crucial resources (e.g., green finance products) brought by digital financial services to improve sustainable performance. Therefore, corporate resilience enables enterprises to utilize digital financial services more efficiently in order to meet diverse needs in the process of sustainable development. Hypothesis 2 proposes that sustainable financial support facilitated by digital finance positively influences corporate resilience, thereby contributing to sustainable development:

**H2a.** *Corporate financial volatility negatively mediates the relationship between digital finance and corporate sustainability performance.*

**H2b.** *Corporate long-term performance growth positively mediates the relationship between digital finance and corporate sustainability performance.*

In summary, the specific role of digital finance in corporate sustainability has been underexplored in the previous literature. While macro-level studies have examined carbon emissions [38,41,46] and micro-level studies have focused on corporate green innovation [2,40,47], the comprehensive impact of digital finance on corporate environmental, social, and governance performance has not been fully investigated. Furthermore, the concept of corporate resilience, which comprises sustainable sales growth and financial stability, is likely to mediate the relationship between digital finance and corporate sustainability performance. Therefore, a more comprehensive empirical analysis of the impact of digital finance on corporate sustainability performance, including the mediating role of corporate resilience, will provide valuable insights into the micro-mechanisms of this process.

## 3. Empirical Strategy

### 3.1. Sample and Data

We utilized a sample of all the listed companies in China's Shanghai Stock Exchange (SHSE) and Shenzhen Stock Exchange (SZSE) from 2011 to 2021 due to the availability of digital-finance-related data from the Institute of Digital Finance of Peking University. A total of 40,890 observations were collected. Special treatment (ST) firms and those with missing or zero values for the main variables were excluded, and tail reduction treatment was conducted for continuous variables at 1% and 99%, resulting in a final total of 23,843 observations representing 2592 enterprises.

Multiple databases were used to construct the sample, including the Institute of Digital Finance of Peking University for digital-finance-index-related data; Sino-Securities Index Information Services (Shanghai, China) Co., Ltd. for environmental, social, and governance (ESG) performance data; and the China Stock Market & Accounting Research Database (CSMAR) and Wind Financial Database for accounting and corporate resilience data. A more detailed breakdown of the information used can be found in Table 2.

**Table 2.** Detailed definition and sources of main variables.

| Variables | Description | Sources |
| --- | --- | --- |
| ESG_index | Enterprises' sustainability performance, measured according to firms' environmental, social, and governance (ESG) performance. | Wind Financial Database |
| DigitFinance | Digital finance, measured using the natural logarithm of the city-level digital finance index. | Institute of Digital Finance of Peking University |
| Volatility | Financial stability, the standard deviation of stock returns for each month in one year. | CSMAR |
| Growth | Sales growth, accumulated sales revenue growth over three years. | CSMAR |

### 3.2. Variable Measurement

1.  **Dependent variable:** Enterprises' sustainability performance (ESG_index). The ESG (environmental, social, and corporate governance) principle (the ESG principle has been developed over 17 years following its formal proposal in 2004) was recently applied to define the sustainability of business activities and measure corporate sustainability performance in an integrated framework [48,49]. Thus, sustainability performance is measured using firms' ESG performance, with values from 1 to 9 (low to high), according to Huazheng's ESG score. (The Huazheng ESG evaluation system provided by Sino-Securities Index Information Services (Shanghai) Co. Ltd. is widely used in China to measure listed companies' ESG performance. Specifically, an enterprise's ESG performance is allocated using 1–9 points, from low to high, according to the Huazheng ESG evaluation system. The higher the value is, the better the ESG performance is.)

2.  **Independent variables:** Digital finance (DigitFinance). According to Guo et al. (2020) [50] and Chen and Zhang (2021) [51], the digital finance index is calculated using the data provided by the Institute of Digital Finance of Peking University.

3.  **Mediators: Corporate resilience**. Corporate resilience refers to a firm's ability to effectively respond to shocks and disruptions [18,52], resulting in improved sales growth and financial stability. Following Ortiz et al. (2016) [28], this paper employs two variables to measure corporate resilience: **financial stability** (Volatility), which is measured using the standard deviation of monthly stock returns within a year, and **sustainable sales growth** (Growth), which is measured using accumulated sales revenue growth over three years.

4.  **Controls:** The natural logarithm of firm total assets (lnsize), state-owned enterprise dummy variables (an indicator variable equals one if a firm is state-owned and zero

otherwise) (*SOE*), research and development expenditures scaled according to book assets (RD), the ratio of the sum of long-term debt and short-term debt to book assets (leverage), the ratio of operating income before depreciation to book assets (ROA), the natural logarithm of city-level fiscal expenditure on science and technology (Techpay), the natural logarithm of city-level GDP per capita (lnperGDP), and the loan balance of financial institutions divided by regional GDP (Findev) are used as control variables in this paper.

*3.3. Model Setting*

To investigate how digital finance affects enterprises' sustainability performance, we firstly utilized two-way fixed effects regression following the Hausman test results ($p < 0.05$) to examine the hypotheses. Considering that digital finance may have a nonlinear relationship with sustainability performance, the square term of the *DigitFinance* index was added. As a result, we formulated the baseline regression model in the following manner:

$$ESG\_index_{i,t} = \beta_0 + \beta_1 DigitFinance_{i,t} + \beta_2 DigitFinance^2_{i,t} + \beta_3 Controls_{i,t} + \psi_{year} + \psi_{firm} + \psi_{industry} + \varepsilon_{i,t} \quad (1)$$

where $ESG\_index_{it}$ represents the sustainability performance of firm *I* in year *t*; $DigitFinance_{it}$ indicates the regional digital finance development level; $DigitFinance^2{}_{it}$ represents the square term of the regional digital finance development level; $Controls_{it}$ represents the control variables; $\psi_{year}$, $\psi_{firm}$, and $\psi_{industry}$ represent the time, individual, and industry fixed effects, respectively; and $\varepsilon_{i,t}$ is the random disturbance term, which satisfies the normal distribution.

In this paper, we applied a mediating effects model to further test H2a–H2b (see Figure 1). We introduced corporate resilience into Equations (2) and (3) to explore how it mediates the relationship between digital finance and corporate sustainability performance.

$$Mediators_{i,t} = \beta_0 + \beta_1 DigitFinance_{i,t} + \beta_2 DigitFinance^2_{i,t} + \beta_3 Controls_{i,t} + \psi_{year} + \psi_{firm} + \psi_{industry} + \varepsilon_{i,t} \quad (2)$$

$$ESG\_index_{i,t} = \beta_0 + \beta_1 DigitFinance_{i,t} + \beta_2 DigitFinance^2_{i,t} + \beta_3 Mediators_{i,t} + \beta_4 Controls_{i,t} + \psi_{year} + \psi_{firm} + \psi_{industry} + \varepsilon_{i,t} \quad (3)$$

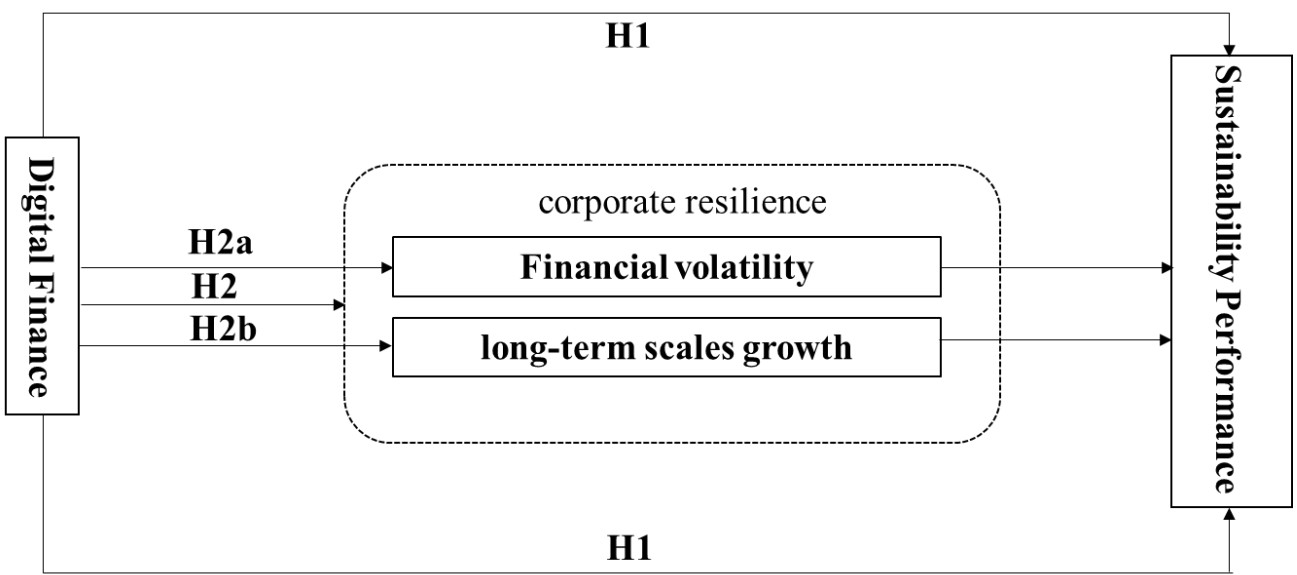

**Figure 1.** Research hypotheses. Source: Author's plot.

Here, $Mediators_{it}$ includes the indicator variables of financial stability (*Volatility*) and sustainable sales growth (*Growth*) that are applied for testing the mediating effect between

digitalization and sustainability performance. Each regression model was subjected to the default robustness standard error procedure.

## 4. Empirical Results

### 4.1. Descriptive Statistics and Correlation Analysis

In this study, empirical tests were conducted using STATA16. Table 3 presents the descriptive results of the related variables along with their variance inflation factor (VIF) values. Corporate sustainability performance (*ESG_index*) has a mean of 4.080 and a standard deviation of 1.141, indicating that the sustainability performance of corporations varies across firms. The mean of digital finance (*DigitFinance*) is 2.335, and its standard deviation is 0.733, suggesting that the digital finance index differs across cities. Corporate resilience has a mean of 18.855 in finance stability (*Volatility*) and 0.002 in sales growth (*Growth*), with standard deviations of 23.310 and 0.006, respectively, indicating that corporate resilience varies significantly between firms. Regarding the control variables, the mean (3.881) and standard deviation (6.919) of the R&D input (*RD*) indicate significant variability in innovation efforts across firms. The means and standard deviations of city-level fiscal expenditure (*Techpay*), GDP per capita (*lnperGDP*), and the finance development level (*Findev*) show significant cross-city variations in the levels of local government intervention in science and technology, economic development, and traditional finance development. Each variable's variance inflation factor (VIF) is much lower than 10, indicating that there is no significant multicollinearity between the variables.

**Table 3.** Description.

| | N | Mean | sd | p25 | Median | p75 | VIF |
|---|---|---|---|---|---|---|---|
| ESG_index | 23,843 | 4.080 | 1.141 | 3.000 | 4.000 | 5.000 | —— |
| DigitFinance | 23,843 | 2.335 | 0.733 | 1.820 | 2.461 | 2.914 | 1.720 |
| Volatility | 23,843 | 18.855 | 23.310 | 5.116 | 10.640 | 22.660 | 1.140 |
| Growth | 23,843 | 0.002 | 0.006 | 0.000 | 0.001 | 0.002 | 1.350 |
| lnSize | 23,843 | 22.063 | 1.340 | 21.094 | 21.869 | 22.812 | 2.080 |
| SOE | 23,843 | 0.352 | 0.478 | 0.000 | 0.000 | 1.000 | 1.170 |
| RD | 23,843 | 3.881 | 6.919 | 0.000 | 0.000 | 0.000 | 1.050 |
| Leverage | 23,843 | 0.442 | 0.216 | 0.268 | 0.431 | 0.603 | 1.650 |
| ROA | 23,843 | 0.036 | 0.061 | 0.013 | 0.036 | 0.067 | 1.230 |
| Techpay | 23,843 | 12.892 | 2.148 | 11.639 | 13.033 | 14.268 | 2.000 |
| lnperGDP | 23,843 | 10.516 | 2.535 | 10.545 | 11.169 | 11.644 | 2.120 |
| Findev | 23,843 | 1.971 | 1.720 | 0.889 | 1.660 | 2.677 | 1.410 |

Notes: all continuous variables are winsorized at the 1st and 99th percentiles.

Table 4 presents both the Pearson correlations and Spearman correlations for the main variables used in this study. The correlation between corporate sustainability performance and digital finance is significantly positive (0.014, 0.038). The correlations between corporate sustainability and the two indicators of corporate resilience are also significantly positive (0.079/0.068, 0.162/0.243). Moreover, the correlations between digital finance and the two indicators of corporate resilience are significantly positive (0.153/0.181, 0.017/0.034). These findings provide preliminary support for the primary research hypothesis that digital finance has a positive impact on both corporate sustainability performance and resilience.

**Table 4.** Correlation.

| | 1 | 2 | 3 | 4 | 5 | 6 | 7 | 8 | 9 | 10 | 11 | 12 |
|---|---|---|---|---|---|---|---|---|---|---|---|---|
| 1. ESG_score | 1.000 | 0.038 *** | 0.068 *** | 0.243 *** | 0.153 *** | 0.067 *** | −0.001 | −0.057 *** | 0.208 *** | 0.054 *** | 0.049 *** | 0.036 *** |
| 2. DigitFinance | 0.014 *** | 1.000 | 0.181 *** | 0.034 *** | 0.108 *** | −0.109 *** | 0.201 *** | −0.027 *** | 0.026 *** | −0.061 *** | 0.618 *** | −0.386 *** |
| 3. Volatility | 0.079 *** | 0.153 *** | 1.000 | 0.173 *** | 0.294 *** | 0.065 *** | 0.134 *** | 0.079 *** | 0.018 *** | 0.040 *** | 0.090 *** | −0.008 |
| 4. Growth | 0.162 *** | 0.017 *** | 0.206 *** | 1.000 | 0.471 *** | 0.038 *** | 0.077 *** | 0.229 *** | 0.259 *** | 0.057 *** | 0.030 *** | 0.052 *** |
| 5. lnSize | 0.178 *** | 0.117 *** | 0.325 *** | 0.497 *** | 1.000 | 0.361 *** | 0.088 *** | 0.479 *** | −0.116 *** | 0.026 *** | 0.120 *** | −0.012 ** |
| 6. SOE | 0.072 *** | −0.109 *** | 0.086 *** | 0.135 *** | 0.371 *** | 1.000 | −0.035 *** | 0.299 *** | −0.188 *** | 0.020 *** | −0.079 *** | 0.065 *** |
| 7. lnRD | 0.005 | 0.190 *** | 0.095 *** | 0.028 *** | 0.072 *** | −0.037 *** | 1.000 | −0.056 *** | 0.013 ** | 0.082 *** | 0.128 *** | 0.023 *** |
| 8. Lev | −0.070 *** | −0.029 *** | 0.100 *** | 0.269 *** | 0.474 *** | 0.300 *** | −0.061 *** | 1.000 | −0.444 *** | −0.013 ** | −0.034 *** | 0.005 |
| 9. ROA | 0.223 *** | −0.016 *** | 0.022 *** | 0.053 *** | −0.041 *** | −0.112 *** | 0.006 | −0.365 *** | 1.000 | 0.060 *** | 0.025 *** | 0.047 *** |
| 10. Techpay | 0.064 *** | −0.104 *** | 0.009 | 0.033 *** | 0.031 *** | 0.013 ** | 0.060 *** | −0.012 ** | 0.066 *** | 1.000 | 0.542 *** | 0.552 *** |
| 11. lnperGDP | 0.041 *** | 0.496 *** | 0.065 *** | 0.029 *** | 0.127 *** | −0.062 *** | 0.115 *** | −0.025 *** | 0.015 *** | 0.377 *** | 1.000 | 0.044 *** |
| 12. Findev | 0.044 *** | −0.290 *** | 0.006 | 0.027 *** | 0.010 * | 0.063 *** | 0.028 *** | 0.003 | 0.058 *** | 0.517 *** | 0.046 *** | 1.000 |

Notes: Significance levels: * $p < 0.1$, ** $p < 0.05$, *** $p < 0.01$. Pearson correlation coefficients are presented below the diagonal, whereas Spearman correlation coefficients are presented above the diagonal.

### 4.2. Baseline Results

Table 5 presents the baseline results of this study. The Hausman test (unreported) suggests that the fixed-effects model is more appropriate. Column (1) displays the basic results of the impact of digital finance on corporate sustainability performance, while column (2) includes control variables. Column (3), on the other hand, comprises the quadratic term of the digital finance and control variables. The regression findings presented in columns (1)–(3) provide robust support for the primary hypothesis (H1). Column (1) reveals that the coefficient for digital finance is significantly negative (coef. = −0.07, t = −10.09), indicating that firms situated in cities with a lower level of digital finance may perform better in sustainability-related behaviors such as environmental, social, and governance performance. In column (2), the coefficient of digital finance is still significantly negative (coef. = −0.22, t = −17.72), even after including the control variables. In column (3), the coefficient of the squared term of digital finance is significantly positive (coef. = 0.00, t = 2.43), proposing a U-shaped relationship between the local digital finance level and corporate sustainability performance.

**Table 5.** Baseline results.

| | (1) | (2) | (3) |
|---|---|---|---|
| | ESG_Score | ESG_Score | ESG_Score |
| DigitFinance | −0.07 *** | −0.22 *** | −0.33 *** |
| | (−10.09) | (−17.72) | (−6.84) |
| DigitFinance2 | | | 0.00 ** |
| | | | (2.43) |
| lnSize | | 0.25 *** | 0.25 *** |
| | | (25.91) | (25.77) |
| lnRD | | −0.00 | −0.00 |
| | | (−0.23) | (−0.38) |
| Lev | | −1.04 *** | −1.04 *** |
| | | (−20.45) | (−20.45) |
| ROA | | 1.42 *** | 1.42 *** |
| | | (11.85) | (11.84) |

**Table 5.** *Cont.*

|  | (1) | (2) | (3) |
|---|---|---|---|
|  | ESG_Score | ESG_Score | ESG_Score |
| Techpay |  | 0.00 | 0.00 |
|  |  | (0.14) | (0.34) |
| lnperGDP |  | 0.08 *** | 0.08 *** |
|  |  | (3.66) | (3.43) |
| Findev |  | −0.02 *** | −0.02 *** |
|  |  | (−3.77) | (−2.58) |
| SOE |  | 0.07 *** | 0.07 *** |
|  |  | (2.65) | (2.58) |
| Year | Yes | Yes | Yes |
| Firm | Yes | Yes | Yes |
| Industry | Yes | Yes | Yes |
| _cons | 4.35 | −1.70 *** | −1.55 *** |
|  | (21.15) | (−6.52) | (−5.87) |
| R2 | 0.07 | 0.171 | 0.172 |
| N | 23,843 | 23,843 | 23,843 |

t statistics in parentheses; ** $p < 0.05$, *** $p < 0.01$.

### 4.3. Mechanism Test and Results

Table 6 displays the mechanism test results of this study. To determine how digital finance shapes corporate sustainability performance, a mediator was introduced to investigate whether corporate resilience underlies the relationship between digital finance and sustainability performance. Columns (1) and (2) show the mechanism test results of enterprise resilience measured according to financial volatility, while columns (3) and (4) report the mechanism test results of enterprise resilience measured according to long-term sales growth performance. The regression results presented in columns (1) and (2) provide robust support for H2a; digital finance has an inverted U-shaped relationship with corporate financial volatility, whereas financial volatility is significantly (negatively) associated with corporate sustainability performance (coef. = −0.00, t = −4.06). This suggests that firms with lower financial volatility perform better in their sustainable practices. The regression findings presented in columns (3) and (4) provide strong support for H2b; digital finance has a U-shaped relationship with corporate sales growth, while sales growth is significantly (positively) associated with corporate sustainability performance (coef. = 6.69, t = 4.88). This indicates that firms with higher sales growth perform better in their sustainable practices. These results confirm that a portion of the mediated effect of corporate resilience is significant, suggesting that corporate resilience improves the allocation efficiency of external resources brought by digital finance, thereby promoting the achievement of multiple corporate goals related to environmental, social, and governance performance.

**Table 6.** Mechanism test and results.

|  | (1) | (2) | (3) | (4) |
|---|---|---|---|---|
|  | Volatility | ESG_Score | Growth | ESG_Score |
| DigitFinance | 20.98 *** | −0.30 *** | −0.00 *** | −0.29 *** |
|  | (18.54) | (−6.18) | (−10.81) | (−6.01) |
| DigitFinance2 | −0.00 *** | 0.00 ** | 0.00 *** | 0.00 * |
|  | (−14.85) | (1.99) | (6.33) | (1.75) |
| Volatility |  | −0.00 *** |  |  |
|  |  | (−4.06) |  |  |
| Growth |  |  |  | 6.69 *** |
|  |  |  |  | (4.88) |

**Table 6.** *Cont.*

|  | (1) | (2) | (3) | (4) |
|---|---|---|---|---|
|  | **Volatility** | **ESG_Score** | **Growth** | **ESG_Score** |
| Control | Included | Included | Included | Included |
| Year | Yes | Yes | Yes | Yes |
| Firm | Yes | Yes | Yes | Yes |
| Industry | Yes | Yes | Yes | Yes |
| _cons | −114.99 *** | −1.93 *** | −0.06 *** | −1.32 *** |
|  | (−19.64) | (−6.09) | (−33.95) | (−4.07) |
| R2 | 0.233 | 0.175 | 0.374 | 0.168 |
| N | 23,843 | 23,843 | 23,843 | 23,843 |

t statistics in parentheses; * $p < 0.1$, ** $p < 0.05$, *** $p < 0.01$.

### 4.4. Heterogeneous Effects

Table 7 presents the regression results of the heterogeneity analysis regarding the firms' scale size, property rights, and regional economic level, respectively. Columns (1) and (2) report the impact of digital finance on sustainability performance across firms of different sizes, indicating that digital finance has a significant, inverted U-shaped relationship with the sustainability performance of small-sized firms (coef. = −0.00, t = −2.59), while it has a significant, U-shaped relationship with that of their larger counterparts (coef. = 0.00, t = 4.03). These findings suggest that the positive effects of digital finance on corporate sustainability performance are more pronounced for small firms, which are typically characterized by stronger financial constraints in the initial stage [18,53]. However, the positive role of digital finance may diminish when external resources are sufficient to meet the sustainable development needs of small firms; thus, more attention should be paid to their financial performance. Columns (3) and (4) report the impact of digital finance on sustainability performance across firms with different property rights, indicating that digital finance has a significant, U-shaped relationship with the sustainability performance of non-state-owned firms (coef. = 0.00, t = 5.42), while it has a significant, negative relationship with their state-owned counterparts (coef. = −0.29, t = −4.87). This reflects the fact that state-owned firms may have more access to government benefits than their counterparts [18], granting digital finance a relatively smaller role in shaping sustainability performance for state-owned firms. Therefore, non-state-owned firms need to focus more on acquiring external financial support to achieve their sustainability goals. Columns (5) and (6) report the impact of digital finance on sustainability performance across firms located in cities with different economic levels, indicating that digital finance has a significant, U-shaped relationship with the sustainability performance of firms situated in cities with a lower economic level (coef. = 0.00, t = 2.40), while it has a significant, negative relationship with their counterparts in cities with a higher level of economic development (coef. = −0.34, t = −4.34). This reflects the fact that firms located in cities with higher economic levels are more likely to access external resources in order to meet multiple corporate development needs, including sustainable development.

**Table 7.** Heterogeneous effects and results.

|  | (1) | (2) | (3) | (4) | (5) | (6) |
|---|---|---|---|---|---|---|
|  | **Scale Size** | | **Property Rights** | | **Economic Levels** | |
|  | **Small Size** | **Large Size** | **State-Owned** | **Non-State-Owned** | **High perGDP** | **Low perGDP** |
| DigitFinance | −0.15 ** | −0.36 *** | −0.29 *** | −0.42 *** | −0.34 *** | −0.19 ** |
|  | (−2.31) | (−6.19) | (−4.87) | (−6.71) | (−4.34) | (−2.54) |
| DigitFinance2 | −0.00 *** | 0.00 *** | −0.00 | 0.00 *** | 0.00 | 0.00 ** |
|  | (−2.59) | (4.03) | (−0.36) | (5.42) | (1.57) | (2.40) |

**Table 7.** *Cont.*

| | (1) | (2) | (3) | (4) | (5) | (6) |
|---|---|---|---|---|---|---|
| | Scale Size | | Property Rights | | Economic Levels | |
| | Small Size | Large Size | State-Owned | Non-State-Owned | High perGDP | Low perGDP |
| Control | Included | Included | Included | Included | Included | Included |
| Year | Yes | Yes | Yes | Yes | Yes | Yes |
| Firm | Yes | Yes | Yes | Yes | Yes | Yes |
| Industry | Yes | Yes | Yes | Yes | Yes | Yes |
| _cons | −0.94 * | −2.26 *** | −0.39 | −2.17 *** | −1.05 *** | −1.14 *** |
| | (−1.92) | (−5.68) | (−1.13) | (−5.16) | (−3.06) | (2.30) |
| R2 | 0.023 | 0.142 | 0.040 | 0.214 | 0.105 | 0.153 |
| N | 11,167 | 14,010 | 9812 | 14,795 | 11,228 | 11,366 |

t statistics in parentheses; * $p < 0.1$, ** $p < 0.05$, *** $p < 0.01$.

### 4.5. Robustness Test

To ensure more robust conclusions, we employed robustness tests and endogenous analysis. Specifically, system GMM estimation was introduced to address the reverse causality relationship between digital finance and corporate sustainability performance. Firms from high-tech industries were excluded to alleviate the potential endogeneity caused by innovation effects. Digital finance, which is highly correlated with both the scale of traditional financial institutions and the level of regional economic development, was substituted with the ratio of the balance of financial institution loans to GDP (loan/GDP) as an alternative measure of digital finance. Moreover, considering that there may be a cubic relationship between regional digital finance and corporate sustainability performance, i.e., an N-shaped or horizontal S-shaped relationship, the cubic form of digital finance was added to the baseline model.

In Table 8, the results in column (1) indicate that the S-GMM regression results of digital finance are consistent with the baseline regression. Furthermore, the results suggest that the baseline conclusion is supported after accounting for endogeneity issues, as the *p*-values of both the Arellano–Bond AR (1) test and AR (2) test are greater than 0.05. In column (2), the coefficient of digital finance is consistent with the baseline regression, implying that the relationship between digital finance and corporate sustainability performance is robust. In column (3), the coefficients of loan/GDP and its quadratic terms are consistent with the baseline regression, suggesting that digital finance shapes sustainable corporate performance. In column (4), the coefficient of the cubic term of digital finance is not significant, indicating that there is no cubic relationship, which confirms the robustness of the U-shaped relationship between digital finance and sustainable corporate performance.

**Table 8.** Endogeneity effects and results.

| | (1) | (2) | (3) | (4) |
|---|---|---|---|---|
| | S-GMM | Excluded High-Tech Samples | Alternative Measures of Digital Finance | The Cubic Form of Digital Finance |
| | ESG_Score | ESG_Score | ESG_Score | ESG_Score |
| L.ESG_score | 0.46 *** | | | |
| | (31.30) | | | |
| DigitFinance | −0.66 *** | −0.32 *** | −0.104 *** | 0.585 *** |
| | (−9.16) | (−6.58) | (−5.12) | (3.17) |
| DigitFinance2 | 0.00 *** | 0.00 ** | 0.01 *** | 0.00 ** |
| | (8.57) | (2.15) | (3.09) | (2.18) |
| DigitFinance3 | | | | 0.093 |
| | | | | (1.14) |

**Table 8.** *Cont.*

| | (1) | (2) | (3) | (4) |
|---|---|---|---|---|
| | **S-GMM** | **Excluded High-Tech Samples** | **Alternative Measures of Digital Finance** | **The Cubic Form of Digital Finance** |
| | **ESG_Score** | **ESG_Score** | **ESG_Score** | **ESG_Score** |
| Control | Included | Included | Included | Included |
| Year | Yes | Yes | Yes | Yes |
| Firm | Yes | Yes | Yes | Yes |
| Industry | Yes | Yes | Yes | Yes |
| _cons | | −1.74 *** | 1.52 *** | −2.10 *** |
| | | (−5.59) | (5.73) | (−6.54) |
| Sargan test/R2 | 67.13 | 0.169 | 0.141 | 0.172 |
| *p*-value | 0.163 | —— | | |
| AR(1): *p*-value | 0.076 | —— | | |
| AR(2): *p*-value | 0.107 | —— | | |
| N | 20,304 | 20,289 | 20,127 | 20,289 |

Notes: t statistics in parentheses; ** $p < 0.05$, *** $p < 0.01$.

### 4.6. Regional Spillover Effects

Digital finance, enabled by advanced digital information technology, has the potential to reduce spatiotemporal barriers and transaction costs, which can accelerate the process of financial agglomeration and diffusion and significantly improve spatial interaction between regions [22]. In addition, this regional spillover effect may affect sustainable corporate performance. Therefore, incorporation of the spillover effects of the digital finance variable (DigitFinance_Ner) into the baseline model can ease this concern. According to Xia et al. (2022) [18], the level of digital finance in cities neighboring that in which the company is registered is a good measure of this spillover effect. To depict this effect, the natural logarithm for the mean value of the loan balance of financial institutions in neighboring cities is used.

The results of the regional spillover effects are reported in Table 9. Column (1) shows the relationship between the spillover effect of digital finance and sustainable corporate performance. It suggests that this effect has a significant impact on sustainable corporate behavior. Moreover, the U-shaped relationship between digital finance and sustainable corporate performance remains robust even after controlling for regional spillover effects, as shown in columns (2) and (3).

**Table 9.** Results of spatial econometric models.

| | (1) | (2) | (3) |
|---|---|---|---|
| | **ESG_Score** | **ESG_Score** | **ESG_Score** |
| DigitFinance | | −0.20 *** | −0.41 *** |
| | | (−5.38) | (−4.70) |
| DigitFinance2 | | | 0.00 *** |
| | | | (3.97) |
| DigitFinance_Ner | 0.42 *** | 0.27 ** | 0.26 ** |
| | (3.94) | (2.56) | (2.40) |
| Control | Included | Included | Included |
| Year | Yes | Yes | Yes |
| Firm | Yes | Yes | Yes |
| Industry | Yes | Yes | Yes |
| _cons | 1.62 *** | −1.40 *** | −1.17 *** |
| | (6.12) | (−4.23) | (−3.48) |
| R2 | 0.139 | 0.144 | 0.145 |
| N | 23,843 | 23,843 | 23,843 |

Notes: t statistics in parentheses; ** $p < 0.05$, *** $p < 0.01$.

## 5. Conclusions and Implications

### 5.1. Conclusions

While a significant amount of the existing literature has focused on enterprises' sustainability performance, the specific roles of digital finance and corporate resilience in this process remain unexplored. This study aimed to address this gap by examining the relationship between digital finance and corporate sustainability, considering the mediating effect of corporate resilience. We empirically tested these relationships using a sample of listed companies from China's Shanghai Stock Exchange (SHSE) and Shenzhen Stock Exchange (SZSE) between 2011 and 2021. The main contribution of this paper is that it empirically analyzed the impact of regional digital finance on the corporate sustainable performance of China's listed companies and its possible micro-mechanisms. We also focused on exploring effective approaches to the heterogeneity, endogeneity, and robustness issues in order to identify the specific relationship between digital finance development and corporate sustainable development. Our findings suggest that digital finance has a U-shaped impact on corporate sustainable performance, indicating that digital finance can significantly promote enterprises' sustainability within a certain range. Specifically, digital finance contributes to the enhancement of sustainability by reducing corporate financial volatility and promoting long-term performance growth. These results imply that companies located in regions with a higher level of digital finance are likely to perform better in both financial and sales terms and can effectively optimize and allocate external resources to promote corporate sustainable transition. In addition, the positive effects of digital finance on sustainability performance are stronger for non-state-owned firms, firms located in cities with lower GDP, and small firms in their initial stages. This finding suggests that digital finance plays a more crucial role in improving the sustainability performance of firms in worse financial conditions with higher business autonomy and better market environments.

### 5.2. Implications

Pro-environmental behavior typically requires a significant amount of resources that can place an additional financial burden on firms. Thus, firms should leverage the advantages of digital finance in alleviating financial constraints and enhancing operational vitality to achieve sustainable development. Additionally, substantial support from the government in regulating financial activities and enhancing financial services is necessary to drive the development of digital finance. In particular, the development of information technology and infrastructure related to 5G, blockchain, and the Internet of Things is essential for promoting regional digital finance. Therefore, the government's substantial support is necessary to drive the development of digital finance. In addition, financial institutions should provide varied financial products and services to meet the financing needs of different enterprises, such as green finance products.

### 5.3. Limitations

While this paper attempted to robustly test our hypotheses, some limitations need to be addressed. The specific role and micro-mechanism of digital finance in corporate sustainability performance were analyzed, but some potential factors associated with corporate sustainability remain unexplored. Future research should investigate other aspects that could influence corporate sustainability in conjunction with digital finance. Additionally, more heterogeneity issues, such as the heterogeneity of space, should be discussed in future studies. Considering regional heterogeneity, it is important to examine the role of digital finance in promoting corporate sustainability in different regions and analyze how this role varies across different regions. Ultimately, addressing these limitations could deepen our understanding of digital finance's contribution to corporate sustainability and inform policymakers about potential approaches to promoting sustainable development efficiently.

**Author Contributions:** Conceptualization, S.H., Q.Z. and X.Z.; methodology, S.H.; software, S.H.; validation, S.H.; formal analysis, S.H.; resources, S.H., Z.X. and Q.Z.; data curation, S.H.; writing—original draft preparation, S.H.; writing—review and editing, Z.X. and Q.Z.; supervision, X.Z. All authors contributed to writing the paper. All authors have read and agreed to the published version of the manuscript.

**Funding:** The authors are grateful for the financial support provided by the Jiangsu Provincial Department of Education Fund of Philosophy and Social Science (2022SJYB1464), Humanities and Social Sciences Foundation of Suzhou University of Science and Technology (XKR202112), and Jiangsu Social Science Applied Research Quality Project (22 SCB-29).

**Institutional Review Board Statement:** Written informed consent was obtained from individual or guardian participants.

**Informed Consent Statement:** Written informed consent for publication was obtained from all participants.

**Data Availability Statement:** The datasets generated during and/or analyses conducted during the current study are available at the http://www.stats.gov.cn/. (accessed on 1 May 2023).

**Acknowledgments:** We acknowledge the support of Chuanming Yang for his comments on earlier drafts of this paper, as well as the seminar participants at Suzhou University of Science and Technology during the initial writing of this paper. We owe special thanks to Yawen Xu for her assistance in the data collection and revision of our work.

**Conflicts of Interest:** The authors declare that they have no known competing financial interests or personal relationships that could have appeared to influence the work reported in this paper.

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
