# Peer review of "Digital Finance and Corporate Sustainability Performance: Promoting or Restricting? Evidence from China’s Listed Companies"

_sustainability, doi:10.3390/su15139855_

Round 1

Reviewer 1 Report

(1)    In order to provide more detailed descriptive statistics, the authors must including the median of the variables used in the analysis.

(2)   Authors should provide the economic significance of the results to highlight their relevance.

(3)   Did the authors test the cubic form of the DigitFinance variable?

(4)   Did the authors take into account the effect of outliers? Are the continuous variables winsorized? Adding a robustness check that talks about the results after winsorizing the continuous variables at the first and 99th percentiles.

(5)   The correlation matrix has to give information on the nature of the correlation (parametric and nonparametric). Ideally, Pearson correlation coefficients have to be presented below the diagonal whereas Spearman correlation coefficients have to be presented above the diagonal.

Author Response

Dear Reviewer 1:

We quite appreciate your favorite consideration and the reviewers’ insightful comments. Now we have revised the manuscript entitled “How digital finance promote corporate sustainability performance? the role of corporate resilience” (ID: sustainability-2386946). We have studied the comments carefully and have made corrections which we hope this revision can make our paper more acceptable. Revised portions are marked in red on the paper. The main corrections in the paper and the response to reviewer’s comments are as follows:

Responds to the reviewer’s comments:

Q1: In order to provide more detailed descriptive statistics, the authors must include the median of the variables used in the analysis

Response: Thanks for your nice suggestion. In fact, in descriptive statistics, we have originally considered the median of the variable and represented it as “p50”. However, we did not explain this in detail in the text, which led to misunderstandings by the reviewers. Therefore, in the newly submitted paper, we have changed “p50” to “Median” and marked it in red so that reviewers can better identify it.

Table 2 Description

N

mean

sd

p25

Median

p75

VIF

ESG_index

23843

4.080

1.141

3.000

4.000

5.000

——

DigitFinance

23843

2.335

0.733

1.820

2.461

2.914

1.720

Volatility

23843

18.855

23.310

5.116

10.640

22.660

1.140

Growth

23843

0.002

0.006

0.000

0.001

0.002

1.350

lnSize

23843

22.063

1.340

21.094

21.869

22.812

2.080

SOE

23843

0.352

0.478

0.000

0.000

1.000

1.170

RD

23843

3.881

6.919

0.000

0.000

0.000

1.050

Leverage

23843

0.442

0.216

0.268

0.431

0.603

1.650

ROA

23843

0.036

0.061

0.013

0.036

0.067

1.230

Techpay

23843

12.892

2.148

11.639

13.033

14.268

2.000

lnperGDP

23843

10.516

2.535

10.545

11.169

11.644

2.120

Findev

23843

1.971

1.720

0.889

1.660

2.677

1.410

Notes: all continuous variables are winsorized at the first and 99th percentiles.

Q2: Authors should provide the economic significance of the results to highlight their relevance.

Response: We deeply appreciate your suggestion. According to your comment, we have added the economic significance of the results to highlight their relevance, as follows:

 “Our findings suggest that there is a U-shaped relationship between digital finance and corporate sustainable performance, indicating that digital finance can significantly promote enterprises' sustainability within a certain range. Furthermore, digital finance contributes to the enhancement of sustainability by reducing corporate financial volatility and promoting long-term performance growth. This result implies that companies located in regions with a higher level of digital finance are likely to perform better on both financial and sales conditions and can effectively optimize and allocate external resources to promote corporate sustainable transition.

We also find that the positive effects of digital finance on sustainability performance are stronger for non-state-owned firms, firms located in cities with lower per GDP, and small firms in their initial stages. This finding suggests that digital finance plays a more crucial role in improving the sustainability performance of firms in worse financial conditions, higher business autonomy, and better market environments.

Overall, this study provides important insights into the ways digital finance and corporate resilience influence firms' sustainability performance. These findings offer practical implications for firms seeking to enhance their sustainable practices through leveraging digital finance and building resilience. ”

Q3: Did the authors test the cubic form of the DigitFinance variable?

Response: We are grateful for your suggestion. According to your suggestion, we have made the following change: We have tested the cubic form of the DigitFinance variable in the robustness test (see Section 4.5 and Table 7). The results (In column (4)) show that the coefficient on the cubic term of digital finance is not significant, indicating that there is no cubic relationship, which confirms the robustness of the U-shaped relationship between digital finance and sustainable corporate performance.

Q4: Did the authors take into account the effect of outliers? Are the continuous variables winsorized? Adding a robustness check that talks about the results after winsorizing the continuous variables at the first and 99th percentiles.

Response: Thank you again for your positive comments. Actually, we have done the work mentioned above but did not annotate it in the paper. Therefore, we have made the following annotations below table 2 to facilitate better identification by you: “Notes: all continuous variables are winsorized at the first and 99th percentiles.”

Q5: The correlation matrix has to give information on the nature of the correlation (parametric and nonparametric). Ideally, Pearson correlation coefficients have to be presented below the diagonal whereas Spearman correlation coefficients have to be presented above the diagonal.

Response: Thank you again for your positive comments. According to your suggestion, we have added Spearman correlation coefficients and made modifications to the correlation matrix.

Table 3 Correlation

1

2

3

4

5

6

7

8

9

10

11

12

1.ESG_score

1.000

0.038***

0.068***

0.243***

0.153***

0.067***

-0.001

-0.057***

0.208***

0.054***

0.049***

0.036***

2.DigitFinance

0.014***

1.000

0.181***

0.034***

0.108***

-0.109***

0.201***

-0.027***

0.026***

-0.061***

0.618***

-0.386***

3. Volatility

0.079***

0.153***

1.000

0.173***

0.294***

0.065***

0.134***

0.079***

0.018***

0.040***

0.090***

-0.008

4. Growth

0.162***

0.017***

0.206***

1.000

0.471***

0.038***

0.077***

0.229***

0.259***

0.057***

0.030***

0.052***

5.lnSize

0.178***

0.117***

0.325***

0.497***

1.000

0.361***

0.088***

0.479***

-0.116***

0.026***

0.120***

-0.012**

6.SOE

0.072***

-0.109***

0.086***

0.135***

0.371***

1.000

-0.035***

0.299***

-0.188***

0.020***

-0.079***

0.065***

7.lnRD

0.005

0.190***

0.095***

0.028***

0.072***

-0.037***

1.000

-0.056***

0.013**

0.082***

0.128***

0.023***

8. Lev

-0.070***

-0.029***

0.100***

0.269***

0.474***

0.300***

-0.061***

1.000

-0.444***

-0.013**

-0.034***

0.005

9. ROA

0.223***

-0.016***

0.022***

0.053***

-0.041***

-0.112***

0.006

-0.365***

1.000

0.060***

0.025***

0.047***

10.Techpay

0.064***

-0.104***

0.009

0.033***

0.031***

0.013**

0.060***

-0.012**

0.066***

1.000

0.542***

0.552***

11.lnperGDP

0.041***

0.496***

0.065***

0.029***

0.127***

-0.062***

0.115***

-0.025***

0.015***

0.377***

1.000

0.044***

12.Findev

0.044***

-0.290***

0.006

0.027***

0.010*

0.063***

0.028***

0.003

0.058***

0.517***

0.046***

1.000

Notes: ①significance levels:* p < 0.1, ** p < 0.05, *** p < 0.01. ②Pearson correlation coefficients are presented below the diagonal, whereas Spearman correlation coefficients are presented above the diagonal.

Reviewer 2 Report

·          

·         Thank you for your work. The study titled “How digital finance promote corporate sustainability performance?” is quite interesting. I am happy to review this piece of research as it has a significant contribution to the body of knowledge. It was an ambitious study.

·         The figure of your theoretical framework» should be edited by highlighting all our hypotheses and their numeration within the conceptual framework. Based on the proposed conceptual model, it’s important to formulate 4 additional hypotheses: The 1st hypothesis for the relation between “Corporate financial volatility” and “corporate sustainability performance” and The 2nd hypothesis for the relation between “Corporate long-term performance growth” and “corporate sustainability performance

·         It is essential to discuss the findings in terms of how they relate to the literature review depicted, in light of what existing theories say. Empirical Analysis and Discussion should be developed separately.

·         Authors should develop research questions and hypotheses to better position and structure the work.

·         There should be more discussion about the population, sample, and sampling technique

·         As your paper investigates the influence of digital finance on corporate sustainability performance based on a sample of all China’s listed companies from the Shanghai Stock Exchange (SHSE) and Shenzhen Stock Exchange (SZSE), it is necessary to perform robustness checks using alternative measurements and models.

·         As your paper investigates Shanghai Stock Exchange (SHSE) and Shenzhen Stock Exchange (SZSE) Database, it is necessary to perform robustness checks using alternative models such as spatial econometrics. In the robustness, you used System-GMM Estimation to address this problem, but I wonder why this methodological approach was not used in the empirical section.

·         The use of OLS instead of GLS, OLS with FE and RE, 2SLS, GMM... should be carefully justified to increase the credibility of your work and empirical approach.

·         Authors are also invited to end up their paper by mentioning not only the main (theoretical, methodological and practical/managerial), but also the limitations of their research and open up other research orientations. Indeed, these parts must be developed separately.

·         English writing ability needs to be strengthened.

Author Response

Dear Reviewer 2:

We quite appreciate your favorite consideration and the reviewers’ insightful comments. Now we have revised the manuscript entitled “How digital finance promote corporate sustainability performance? the role of corporate resilience” (ID: sustainability-2386946). We have studied the comments carefully and have made corrections which we hope this revision can make our paper more acceptable. Revised portions are marked in red on the paper. The main corrections in the paper and the response to reviewer’s comments are as follows:

Responds to the reviewer’s comments:

Q1: The figure of your theoretical framework should be edited by highlighting all our hypotheses and their numeration within the conceptual framework. Based on the proposed conceptual model, it’s important to formulate 4 additional hypotheses: The 1st hypothesis for the relation between “Corporate financial volatility” and “corporate sustainability performance” and The 2nd hypothesis for the relation between “Corporate long-term performance growth” and “corporate sustainability performance”.

Response: Thanks for your nice suggestions. Since a flood of literature have explored the relation between “Corporate financial volatility” and “corporate sustainability performance” (Briguglio et al., 2009; Denrell et al., 2013;Markman and Venzin, 2014)[][][] as well as the relation between “Corporate long-term performance growth” and “corporate sustainability performance” (Sharma et al., 2006; Jian and Binghan, 2018; Xiu'e et al., 2021; Man et al., 2022)[][][][], more attention are thereby attached to analyze the relation between digital finance and corporate resilience in our paper.

Q2: It is essential to discuss the findings in terms of how they relate to the literature review depicted, in light of what existing theories say.

Response: We deeply appreciate your suggestion. According to your suggestion, we have made some modifications as follows: “Our findings suggest that there is a U-shaped relationship between digital finance and corporate sustainable performance, indicating that digital finance can significantly promote enterprises' sustainability within a certain range. Furthermore, digital finance contributes to the enhancement of sustainability by reducing corporate financial volatility and promoting long-term performance growth. This result is consistent with Liu et al. (2021) and Li et al. (2021), implies that companies located in regions with a higher level of digital finance are likely to perform better on both financial and sales conditions and can effectively optimize and allocate external resources to promote corporate sustainable transition.”

Q3: Empirical Analysis and Discussion should be developed separately.

Response: We are grateful for your suggestion. According to your suggestion, we have made some modifications as follows:

“5. Conclusions and implications

5.1 Conclusions

While a significant amount of existing literature has focused on enterprises' sustainability performance, the specific roles of digital finance and corporate resilience in this process remain unexplored. This study aims to address this gap by examining the relationship between digital finance and corporate sustainability, considering the mediating effect of corporate resilience. We empirically test these relationships using a sample of listed companies from China's Shanghai Stock Exchange (SHSE) and Shenzhen Stock Exchange (SZSE) between 2011 and 2021.

Our findings suggest that there is a U-shaped relationship between digital finance and corporate sustainable performance, indicating that digital finance can significantly promote enterprises' sustainability within a certain range. Furthermore, digital finance contributes to the enhancement of sustainability by reducing corporate financial volatility and promoting long-term performance growth. This result is consistent with Liu et al. (2021) and Li et al. (2021), implies that companies located in regions with a higher level of digital finance are likely to perform better on both financial and sales conditions and can effectively optimize and allocate external resources to promote corporate sustainable transition.

We also find that the positive effects of digital finance on sustainability performance are stronger for non-state-owned firms, firms located in cities with lower per GDP, and small firms in their initial stages. This finding suggests that digital finance plays a more crucial role in improving the sustainability performance of firms in worse financial conditions, higher business autonomy, and better market environments.

Overall, this study provides important insights into the ways digital finance and corporate resilience influence firms' sustainability performance. These findings offer practical implications for firms seeking to enhance their sustainable practices through leveraging digital finance and building resilience.  

5.2 Implications

The study's findings have important policy implications for promoting sustainable development and leveraging digital finance. Firstly, firms must take advantage of digital finance to achieve sustainable development. Pro-environmental behavior requires significant resources that can put an additional financial burden on firms. Digital finance can alleviate financing constraints, enhance operational vitality, and promote sustainable development for enterprises.

Secondly, it is recommended that policies and measures on digital finance are improved to regulate financial activities and enhance financial services. The development of information technology and infrastructure related to 5G, blockchain, and the Internet of Things is essential for promoting regional digital finance. Therefore, the government's substantial support is necessary to drive the development of digital finance.

Thirdly, differentiated financial service strategies should be developed for enterprises of different scales, property attributes, and regions. Small and medium-sized enterprises could be provided with a variety of financial products specially designed for their needs, while regions with lower levels of economic development require perfecting digital infrastructure. By implementing these strategies, policy-makers can promote digital finance's benefits while ensuring sustainable and equitable corporate development.

5.3 Limitations

While this paper has attempted to robustly test the hypotheses, some limitations need to be addressed. Firstly, while the specific role and micro mechanism of digital finance in corporate sustainability performance are analyzed, some potential factors associated with corporate sustainability remain unexplored. Future research should investigate other aspects that could influence corporate sustainability in conjunction with digital finance.

Secondly, this study measures the resilience of enterprises solely from the output perspective, namely financial volatility and sales growth. Further research should explore this topic from alternative aspects such as characteristics, processes, and capabilities, among others, to provide a more complete picture of corporate resilience.

Lastly, more heterogeneity issues, such as the heterogeneity of space, should be discussed in future studies. Considering the regional heterogeneity, it is important to examine the role of digital finance in promoting corporate sustainability in different regions and analyze how this role varies across different regions. Ultimately, addressing these limitations could deepen our understanding of digital finance's contribution to corporate sustainability and inform policymakers about potential approaches to promoting sustainable development efficiently.”

Q4: Authors should develop research questions and hypotheses to better position and structure the work.

Response: Thanks for your nice suggestions. We have developed our research hypotheses, as follows:

“H1. There is a U-shaped relationship between digital finance and corporate sustainability performance. H2a. Corporate financial volatility negatively mediates the relation-ship between digital finance and corporate sustainability performance. H2b. Corporate long-term performance growth positively mediates the relationship between digital finance and corporate sustainability performance.”

Q5: There should be more discussion about the population, sample, and sampling technique

Response: Thanks for your nice suggestions again. The population, sample, and sampling techniques have been elaborated in detail in the new manuscript. Given the limited length, we cannot delve too deeply in this part. Of course, if the reviewer deems it necessary, we will further supplement the explanation in this section

Q6: As your paper investigates the influence of digital finance on corporate sustainability performance based on a sample of all China’s listed companies from the Shanghai Stock Exchange (SHSE) and Shenzhen Stock Exchange (SZSE), it is necessary to perform robustness checks using alternative measurements and models.

Response: We deeply appreciate your suggestion. According to your suggestion, we have added a robustness test using alternative measurements, as follows:1) digital finance is considered to be highly correlated with both the scale of traditional financial institutions and the levels of regional economic development, the ratio of the balance of financial institution loan to GDP (loan/GDP) is applied as the alternative measures of digital finance; 2) considering that there may be a cubic relationship between regional digital finance and corporate sustainability performance, namely an N-shaped or horizontal S-shaped relationship, the cubic form of the digital finance is added in the baseline model. The results are reported in table 7.

Table 7 Endogeneity effects and results

(1)

(2)

(3)

(4)

S-GMM

Excluded High-tech samples

 Alternative measures of digital finance

The cubic form of the digital finance

ESG_score

ESG_score

ESG_score

ESG_score

L.ESG_score

0.46***

(31.30)

DigitFinance

-0.66***

-0.32***

-0.104***

0.585***

(-9.16)

(-6.58)

(-5.12)

(3.17)

DigitFinance2

0.00***

0.00**

0.01***

0.00**

(8.57)

(2.15)

(3.09)

(2.18)

DigitFinance3

0.093

(1.14)

Control

Included

Included

Included

Included

Year

Yes

Yes

Yes

Yes

Firm

Yes

Yes

Yes

Yes

Industry

Yes

Yes

Yes

Yes

_cons

-1.74***

1.52***

-2.10***

(-5.59)

(5.73)

(-6.54)

Sargan test/R2

67.13

0.169

0.141

0.172

P-value

0.163

——

AR(1):P-value

0.076

——

AR(2):P-value

0.107

——

N

20304

20289

20127

20289

Notes: t statistics in parentheses; * p < 0.1, ** p < 0.05, *** p < 0.01

Q7: As your paper investigates Shanghai Stock Exchange (SHSE) and Shenzhen Stock Exchange (SZSE) Database, it is necessary to perform robustness checks using alternative models such as spatial econometrics. In the robustness, you used System-GMM Estimation to address this problem, but I wonder why this methodological approach was not used in the empirical section.

Response: We deeply appreciate your suggestions again.

  • it is necessary to consider the spatial spillover effects of digital finance because of its inherent spatial spillover characteristics. However, a spatial econometric model to be used is limited by its data characteristics. Since our data is a non-equilibrium panel, a spatial econometric model cannot be used in this paper (spatial econometric models only apply to equilibrium panels). As such, we refer to previous literature that uses the level of digital finance in neighboring cities where the company is registered as the control variables to catch this spatial spillover effect. The results are reported in table 8.
  • in general, replacing regression methods is a commonly used robustness test. The baseline regression uses a two-stage least squares estimation method (The GMM model is mainly suitable for scenarios with heteroscedasticity assumption and significant sequence correlation, where its estimation results are more effective). We propose to replace the different measures in the robustness test process, i.e., we use the GMM method for robustness testing. Of course, we can put the CMM estimation results into the baseline regression model if the reviewer feels necessary.

Table 8 Results of spatial econometric models

(1)

(2)

(3)

ESG_score

ESG_score

ESG_score

DigitFinance

-0.20***

-0.41***

(-5.38)

(-4.70)

DigitFinance2

0.00***

(3.97)

DigitFinance_Ner

0.42***

0.27**

0.26**

(3.94)

(2.56)

(2.40)

Control

Included

Included

Included

Year

Yes

Yes

Yes

Firm

Yes

Yes

Yes

Industry

Yes

Yes

Yes

_cons

1.62***

-1.40***

-1.17***

(6.12)

(-4.23)

(-3.48)

R2

0.139

0.144

0.145

N

23843

23843

23843

Notes: t statistics in parentheses; * p < 0.1, ** p < 0.05, *** p < 0.01

Q8: The use of OLS instead of GLS, OLS with FE and RE, 2SLS, GMM... should be carefully justified to increase the credibility of your work and empirical approach.

Response: Thanks for your nice suggestions. Given the characteristics of the data, we chose OLS with FE as the baseline model. Considering that this model may not robust due to the heteroscedasticity in the model or sequence correlation in the data, as well as potential endogeneity, we use the substitution model and instrumental variables method to ensure the robustness of baseline regression.

Q9: Authors are also invited to end up their paper by mentioning not only the main (theoretical, methodological and practical/managerial), but also the limitations of their research and open up other research orientations. Indeed, these parts must be developed separately.

Response: We are grateful for your suggestion. the limitation is added as follows:" 5.3 Limitations. Although this paper has attempted to robustly test these hypotheses, some limitations remain. Firstly, the specific role and micro mechanism of digital finance in corporate sustainability performance are systematically analyzed in this paper, but there are still some potential factors associated with corporate sustainability. Secondly, the resilience of enterprises is measured solely from the output respect, namely financial volatility and sales growth, so further research should investigate this topic from alternative aspects such as characteristics, processes, capabilities and so on. In addition, more heterogeneity issues, such as space heterogeneity, should be discussed in future studies.”

Q10: English writing ability needs to be strengthened.

Response: Thank you again for your positive comments. We have c revised the grammar, punctuation, and conciseness of the paper, and had proofread it through third-party institutions. Detail information can be seen in the new manuscript.

Reviewer 3 Report

Accept the manuscript without any comments.

Author Response

Dear Reviewer 3:

We quite appreciate your favorite consideration and the reviewers’ insightful comments. Now we have revised the manuscript entitled “How digital finance promote corporate sustainability performance? the role of corporate resilience” (ID: sustainability-2386946). We have studied the comments carefully and have made corrections which we hope this revision can make our paper more acceptable. Revised portions are marked in red on the paper.

Special thanks to reviewer #3 for your good comments. 

Reviewer 4 Report

sustainability-2386946-peer-review-v1
How digital finance promote corporate sustainability performance? The
role of corporate resilience.

In this paper, author shares a study on digital finance in leveraging corporate sustainability 8 performance by analysing the mediating effect of corporate resilience. Although the subject is interesting, I cannot accept the paper in the current form due to the following reasons:

1.             The title is not clear, appealing, interesting and specific. I suggest to revise the paper title to make it more concise and suitable.

2.             The "Where" below Eq. (1) should be "where". Remove the similar problems in your paper.

3.             Abstract should have one sentence per each: context and background, motivation, hypothesis, methods, results, conclusions. What problem did you study and why is it important? What methods did you use? What were your main results? And what conclusions can you draw from your results? Please make your abstract with more specific and quantitative results while it suits broader audiences.

4.             Equations (1)- (3) are not typed in correct form. There is a typing mistake. Author is suggested to look all these typing mistakes.

5.             In general, all variables and Greek letters should be in italic format, and all constants should not be in italic format. Vectors or matrix variables should be in bold and italic format. Please double check the equations used in the manuscript.

6.             The authors should correct the punctuations throughout the manuscript. For example at the end of equations (1), (2),..., put punctuation.

7.             Novelty not clear. A comprehensive table for literature survey should be presented by the authors to show the literature review based on their assumptions, methods, and results.

8.             The literature review is very lengthy. As such, I suggest the author reduces this section to keep only the most important elements.

9.             The references are not formatted. It reads like a mess. The references cited in the text don’t correspond to the references themselves. When this list gets too long, readers lose interest to read further. In addition, complete all the references with their missing information such as vol., issue, page number, year, etc.

10.         There are many grammar errors, English language errors on every page and paragraph. The author is suggested to revise the paper to remove these errors.

11.         Conclusion is very long. It should be decreased. Finally, the main questions should be answered in conclusion section: a) who needs this, b) what is the contribution of your paper, c) what benefit have investors if they decide to use the proposed approach in their digital finance.

***

1.           There are many grammar errors, English language errors on every page and paragraph. The author is suggested to revise the paper to remove these errors.

Author Response

Dear Reviewer 4:

We quite appreciate your favorite consideration and the reviewers’ insightful comments. Now we have revised the manuscript entitled “How digital finance promote corporate sustainability performance? the role of corporate resilience” (ID: sustainability-2386946). We have studied the comments carefully and have made corrections which we hope this revision can make our paper more acceptable. Revised portions are marked in red on the paper.

Responds to the reviewer’s comments:

Q1: The title is not clear, appealing, interesting and specific. I suggest to revise the paper title to make it more concise and suitable.

Response: Thanks for your nice suggestion. According to your suggestion, we have made the following modifications to the title: “Digital finance and corporate sustainability performance: promoting or restricting? Evidence from China’s listed companies”.

Q2: The "Where" below Eq. (1) should be "where". Remove the similar problems in your paper.

Response: We deeply appreciate your suggestion. According to your suggestion, we have corrected it and removed similar problems in our paper. The revised part has been marked in red.

Q3: Abstract should have one sentence per each: context and background, motivation, hypothesis, methods, results, conclusions. What problem did you study and why is it important? What methods did you use? What were your main results? And what conclusions can you draw from your results? Please make your abstract with more specific and quantitative results while it suits broader audiences.

Response: Thanks for your nice suggestion. According to your suggestion, we have made the following modifications to the abstract:” The rapid development of digital finance has optimized the allocation of credit resources, which is of great significance for a sustainable corporate transition. However, little is known about whether and to what extent digital finance affects this process. This paper focuses on the role of digital finance in leveraging corporate sustainability per-formance by using the two-way Fixed Effect Model and Mediating Effect Model to explore the impact of digital finance on green enter-prise innovation and its mechanism. The findings suggest: i) the im-pact of digital finance on the sustainable performance of enterprises presents a U-shaped (coef. =0.00, t=2.43) pattern of first restricting and then promoting, and this conclusion remains robust after considering endogeneity;  ii) the mechanism analysis indicated that digital fi-nance enhances sustainability performance by suppressing corporate financial volatility (coef. =-0.00, t=−4.06) and promoting long-term performance growth (coef. =6.69, t=4.88); iii) the positive effects of digital finance on sustainability performance are more significant for non-state-owned firms (coef. =0.00, t =5.42), firms located in cities with lower per GDP (coef. =0.00, t =2.40), and small firms (coef. =-0.00, t = −2.59) in their initial stage.”

Q4: Equations (1)- (3) are not typed in the correct form. There is a typing mistake. Author is suggested to look all these typing mistakes.

Response: We deeply appreciate your suggestion. According to your suggestion, we have retyped the equations(1)-(3) again in the correct form. And they are marked in red, on the paper.

Q5: In general, all variables and Greek letters should be in italic format, and all constants should not be in italic format. Vectors or matrix variables should be in bold and italic format. Please double check the equations used in the manuscript.

Response: Thanks for your nice suggestion again. According to your suggestion, we have reviewed all variables and corrected them that are not in a right form, and marked in red in the new manuscript.

Q6: The authors should correct the punctuations throughout the manuscript. For example at the end of equations (1), (2),..., put punctuation.

Response: Thanks for your nice suggestion again. According to your suggestion, we have make the revision in the new manuscript.

Q7: Novelty not clear. A comprehensive table for literature survey should be presented by the authors to show the literature review based on their assumptions, methods, and results.

Response: We deeply appreciate your suggestion. For the novelty of our research, we have provided a more detailed explanation to make it clear.

“This study might contribute to present literature in several ways: first, previous studies mainly focus on analyzing the influencing factors of corporate sustainable development from environmental regulation, government subsidies, and technical advancement [50,51]. As an emerging financial service model, digital finance is recognized to significantly affect firms’ diversified goals towards sustainable transition [52]. A novel perspective, regional digital finance, is thereby applied in this paper to explore the specific approach to promote corporate sustainability performance. Second, little is known about whether and to what extent digital finance affects corporate sustainability performance. This study riches previous literature by providing empirical evidence for the nonlinear impact of digital finance on sustainability performance. Meanwhile, the mediating role of corporate resilience offers a new angle on revealing the “black box” of the theoretical relationship between digital finance and corporate sustainable development. Third, several methods are applied to deal with endogeneity and the robustness of the baseline model. ”

Q8: The literature review is very lengthy. As such, I suggest the author reduces this section to keep only the most important elements.

Response: Thanks for your nice suggestion again. According to your suggestion, we have reduced this section to keep only the most important elements. The revised part can be seen in the marked manuscript.

Q9: The references are not formatted. It reads like a mess. The references cited in the text don’t correspond to the references themselves. When this list gets too long, readers lose interest to read further. In addition, complete all the references with their missing information such as vol., issue, page number, year, etc.

Response: We deeply appreciate your suggestion again. We have checked all references to ensure that the citations in the text correspond with the references themselves. Additionally, we have corrected the format of all references according to the journal's requirements. The revised part  can be seen in the new manuscript.

Q10:There are many grammar errors, English language errors on every page and paragraph. The author is suggested to revise the paper to remove these errors.

Response: Thank you again for your positive comments. We have c revised the grammar, punctuation, and conciseness of the paper, and had proofread it through third-party institutions. Detail information can be seen in the new manuscript.

Q11: Conclusion is very long. It should be decreased. Finally, the main questions should be answered in conclusion section: a) who needs this, b) what is the contribution of your paper, c) what benefit have investors if they decide to use the proposed approach in their digital finance.

Response: Thanks for your nice suggestion. According to your suggestion, we have made the following modifications to the conclusion part.

“5. Conclusions and implications

5.1 Conclusions

While a significant amount of existing literature has focused on enterprises' sustainability performance, the specific roles of digital finance and corporate resilience in this process remain unexplored. This study aims to address this gap by examining the relationship between digital finance and corporate sustainability, considering the mediating effect of corporate resilience. We empirically test these relationships using a sample of listed companies from China's Shanghai Stock Exchange (SHSE) and Shenzhen Stock Exchange (SZSE) between 2011 and 2021.

Our findings suggest that there is a U-shaped relationship between digital finance and corporate sustainable performance, indicating that digital finance can significantly promote enterprises' sustainability within a certain range. Furthermore, digital finance contributes to the enhancement of sustainability by reducing corporate financial volatility and promoting long-term performance growth. This result is consistent with Liu et al. (2021) and Li et al. (2021), implies that companies located in regions with a higher level of digital finance are likely to perform better on both financial and sales conditions and can effectively optimize and allocate external resources to promote corporate sustainable transition.

We also find that the positive effects of digital finance on sustainability performance are stronger for non-state-owned firms, firms located in cities with lower per GDP, and small firms in their initial stages. This finding suggests that digital finance plays a more crucial role in improving the sustainability performance of firms in worse financial conditions, higher business autonomy, and better market environments.

Overall, this study provides important insights into the ways digital finance and corporate resilience influence firms' sustainability performance. These findings offer practical implications for firms seeking to enhance their sustainable practices through leveraging digital finance and building resilience.  

5.2 Implications

The study's findings have important policy implications for promoting sustainable development and leveraging digital finance. Firstly, firms must take advantage of digital finance to achieve sustainable development. Pro-environmental behavior requires significant resources that can put an additional financial burden on firms. Digital finance can alleviate financing constraints, enhance operational vitality, and promote sustainable development for enterprises.

Secondly, it is recommended that policies and measures on digital finance are improved to regulate financial activities and enhance financial services. The development of information technology and infrastructure related to 5G, blockchain, and the Internet of Things is essential for promoting regional digital finance. Therefore, the government's substantial support is necessary to drive the development of digital finance.

Thirdly, differentiated financial service strategies should be developed for enterprises of different scales, property attributes, and regions. Small and medium-sized enterprises could be provided with a variety of financial products specially designed for their needs, while regions with lower levels of economic development require perfecting digital infrastructure. By implementing these strategies, policy-makers can promote digital finance's benefits while ensuring sustainable and equitable corporate development.

5.3 Limitations

While this paper has attempted to robustly test the hypotheses, some limitations need to be addressed. Firstly, while the specific role and micro mechanism of digital finance in corporate sustainability performance are analyzed, some potential factors associated with corporate sustainability remain unexplored. Future research should investigate other aspects that could influence corporate sustainability in conjunction with digital finance.

Secondly, this study measures the resilience of enterprises solely from the output perspective, namely financial volatility and sales growth. Further research should explore this topic from alternative aspects such as characteristics, processes, and capabilities, among others, to provide a more complete picture of corporate resilience.

Lastly, more heterogeneity issues, such as the heterogeneity of space, should be discussed in future studies. Considering the regional heterogeneity, it is important to examine the role of digital finance in promoting corporate sustainability in different regions and analyze how this role varies across different regions. Ultimately, addressing these limitations could deepen our understanding of digital finance's contribution to corporate sustainability and inform policymakers about potential approaches to promoting sustainable development efficiently.”

Round 2

Reviewer 2 Report

§   Thank you for revising the paper. Even though the paper contains some changes regarding the previous review results, there is still room for improvement.

§  The paper still needs some language editing and corrections.

Author Response

Thank you again for your positive comments. We have revised the grammar, punctuation, and conciseness of the paper, and had proofread it through third-party institutions, namely, English language editing by MDPI. Detail information can be seen in the new manuscript.

Reviewer 4 Report

The paper has not been improved up to mark. In addition, the author’s responses are not satisfactory. 

The authors have made significant adjustments to the initial submission and I commend them for their efforts. However, they should realize that when a reviewer presents authors with queries they must answer or rebut such queries. In this case, the authors have not answered all the queries. Let me reiterate the unanswered queries:

1.      Non-inclusion of a comprehensive table for literature survey. I queried the non-inclusion of a comprehensive table for literature survey and the authors indicated that they have done it in section 1. However, when I see in chapter 1, it is not there. In this case, these theories do not appear as frameworks.

2.      The authors claimed that they have edited the work sufficiently, but the work is still fraught with numerous grammatical errors. The author is suggested to revise the paper by native speaker.

3.      Abstract should have one sentence per each: context and background, motivation, hypothesis, methods, results, conclusions. What problem did you study and why is it important? What methods did you use? What were your main results? And what conclusions can you draw from your results? Please make your abstract with more specific and quantitative results while it suits broader audiences.

4.      Equations (2)- (3) are not typed in the correct form. There is a typing mistake. Author is suggested to look at all these typing mistakes.

5.      It is hard to understand the novelty of this study with the current form of introduction. That is why, I suggest to include a literature matrix to show main novelty parts of the study.

6.      The references are not formatted. It reads like a mess. The references cited in the text don’t correspond to the references themselves. When this list gets too long, readers lose interest to read further. In addition, complete all the references with their missing information such as vol., issue, page number, year, etc.

7.      The current form of Conclusion is very long. It should be decreased. Finally, the main questions should be answered in the conclusion section: a) who needs this, b) what is the contribution of your paper, c) what benefit have investors if they decide to use the proposed approach in their digital finance.

8.      The methodology section is weak. Author should clearly mention the methodology and reasons for using it. which is missing.

Extensive editing of English language required

Author Response

Dear Reviewer 4: We quite appreciate your favorite consideration and the reviewers’ insightful comments. Now we have revised the manuscript entitled “Digital finance and corporate sustainability performance:promoting or restricting? Evidence from China’s listed companies ” (ID: sustainability-2386946). We have studied the comments carefully and have made corrections which we hope this revision can make our paper more acceptable. Revised portions are marked in red on the paper.

Responds to the reviewer’s comments:

Q1: Non-inclusion of a comprehensive table for literature survey. I queried the non-inclusion of a comprehensive table for literature survey and the authors indicated that they have done it in section 1. However, when I see in chapter 1, it is not there. In this case, these theories do not appear as frameworks.

Response: We are really sorry about our mistakes, and deeply appreciate your suggestion. We have provided a comprehensive table for overview of contributions to the topic as follow: Table 1 Overview of contributions to the topic Author(s) Assumption Method Empirical result Broby et al., (2018) The joint collaboration in fintech will benefit the output and labour productivity. Case Analysis; Theoretic Deduction New fintech methods and innovations positively increase shareholder value, productivity, and employment. Li and Liu, (2021) The digital inclusive finance can indirectly promote the development of green economy by increasing the concentration of producer services and optimizing the upgrading of industrial structure. Two-way Fixed Effects Model; Threshold Effect mode; instrumental variable model (IV); SYS-GMM There is a significant positive U-shaped nonlinear relationship between digital inclusive finance and green development. Liu et al., (2022) Digital financial inclusion is able to promote China's economic growth through promoting small and mediumsized enterprise entrepreneurship and stimulating residents' consumption. VAR model; Threshold Regression Model; Multiple Intermediary Models. The impact of digital financial inclusion development on economic growth has a significant Internet threshold effect. Cao et al., (2021) Green technology innovation is the transmission path through which digital finance affects energy-environmental performance. Two-way Fixed Effects Model; Mediating Effect Model; Difference in Difference (DID) Model Digital finance significantly improves China’s energy-environmental performance,and green-tech innovation plays an important intermediary role in this process. Wang et al., (2022). Digital financial inclusion may affect carbon emissions between cities through its breadth of coverage, depth of use, and level of digitization. Spatial econometric model; Mediating Effect Model Digital financial inclusion positively impacts CO2 emissions of local cities, but negatively impacts neighboring cities. Lin and Ma, (2022) Digital finance can significantly enhance the efficiency of capital allocation and reduce the financing cost, which is beneficial to alleviate the financing constraints of green innovation Two-way Fixed Effects Model; Mediating Effect Model Digital finance can improve the quantity and quality of green technological innovation; digital finance indirectly improves green innovation mainly by alleviating financing constraints. Zhang and Liu, (2022). Digital finance can help solve the funding dilemma of geen-tech innovation and combines well with carbon emission efficiency for a better carbon emission effificiency. Two-way Fixed Effects Model; Spatial econometric model The synergistic effect of digital finance and green technological innovation plays a significant role in promoting local carbon emission efficiency but suppresses carbon emission efficiency in surrounding cities to some extent.

Q2: The authors claimed that they have edited the work sufficiently, but the work is still fraught with numerous grammatical errors. The author is suggested to revise the paper by native speaker.

Response: Thank you again for your positive comments. We have revised the grammar, punctuation, and conciseness of the paper, and had proofread it through third-party institutions, namely, English language editing by MDPI. Detail information can be seen in the new manuscript.

Q3: Abstract should have one sentence per each: context and background, motivation, hypothesis, methods, results, conclusions. What problem did you study and why is it important? What methods did you use? What were your main results? And what conclusions can you draw from your results? Please make your abstract with more specific and quantitative results while it suits broader audiences.

Response: Thanks for your nice suggestion. According to your suggestion, we have made the following modifications to the abstract: Context and background: “The development of internet platforms and information technology has accelerated the transformation of conventional finance. ” Motivation: “However, whether, and to what extent, digital finance empirically affects this process is still not well understood.” Hypothesis: “Emerging digital finance is expected to optimize the allocation of credit resources and thereby promote a sustainable transition for corporations. ” Methods: “This paper investigates the role of digital finance in promoting corporate sustainability performance by exploring its impact on green enterprise innovation and its mechanism using the two-way fixed effect model and the mediating effect model. ” Results: “The findings suggest the following: i) The impact of digital finance on the sustainable performance of enterprises follows a U-shaped (coef. =0.00, t=2.43) pattern, where digital finance initially restricts and then promotes the sustainable performance of enterprises. This conclusion remains robust even after considering endogeneity. ii) The mechanism analysis indicates that digital finance enhances sustainability performance by reducing corporate financial volatility (coef. =-0.00, t=−4.06) and promoting long-term performance growth (coef. =6.69, t=4.88). iii) The positive effects of digital finance on sustainability performance are more significant for non-state-owned firms (coef. =0.00, t =5.42), firms located in cities with a lower GDP per capita (coef. =0.00, t =2.40), and smaller firms (coef. =-0.00, t = −2.59) in their initial stages. ” Conclusions: “These results imply that China should accelerate digitization in the financial markets, and thus further develop its potentials in sustainable development. ”

Q4: Equations (2)- (3) are not typed in the correct form. There is a typing mistake. Author is suggested to look at all these typing mistakes.

Response: We deeply appreciate your suggestion. According to your suggestion, we have retyped the equations(1)-(3) again in the correct form as follow:

Q5: It is hard to understand the novelty of this study with the current form of introduction. That is why, I suggest to include a literature matrix to show main novelty parts of the study.

Response: We deeply appreciate your suggestion. Table 1 clearly shows that previous literature on this topic mainly focus on the sustainable effects and mechanism of digital finance at the macro level. As such, our study aims to address this gap by examining the relationship between digital finance and corporate sustainability, considering the mediating effect of corporate resilience, which contribute to present literature at the micro level. Meanwhile, the sustainable development measured by ESG index encompasses multiple dimensions of sustainable development characteristics such as environment, society, and governance, which is more comprehensive than analyzing solely from the environmental aspect. Also, the mediating role of corporate resilience offers a new angle revealing the “black box” of the theoretical relationship between digital finance and corporate sustainable development. In addition, several methods are applied to deal with endogeneity and the robustness of the baseline model.

Q6: The references are not formatted. It reads like a mess. The references cited in the text don’t correspond to the references themselves. When this list gets too long, readers lose interest to read further. In addition, complete all the references with their missing information such as vol., issue, page number, year, etc.

Response: We are really sorry about our mistakes, and deeply appreciate your suggestion again. We have checked all references to ensure that the citations in the text correspond with the references themselves. Additionally, we have corrected the format of all references according to the journal's requirements. The revised part can be seen in the new manuscript.

Q7: The current form of the Conclusion is very long. It should be decreased. Finally, the main questions should be answered in the conclusion section: a) who needs this, b) what is the contribution of your paper, c) what benefit have investors if they decide to use the proposed approach in their digital finance.

Response: Thanks for your nice suggestion again. According to your suggestion, we have made the revision in the new manuscript. a)who needs this: “firms should leverage the advantages of digital finance in alleviating financial constraints and enhancing operational vitality to achieve sustainable development. Additionally, substantial support from the government in regulating financial activities and enhancing financial services is necessary to drive the development of digital finance. In particular, the development of information technology and infrastructure related to 5G, blockchain, and the Internet of Things is essential for promoting regional digital finance. Therefore, the government's substantial support is necessary to drive the development of digital finance. In addition, financial institutions should provide varied financial products and services to meet the financing needs of different enterprises, such as green finance products.” b)what is the contribution of your paper: “The main contribution of this paper is to empirically analyze the impact of regional digital finance on corporate sustainable performance of China's listed companies and its possible micro mechanisms. We also focused on exploring effective approaches to the heterogeneity, endogeneity and robustness issues to identify the specific relationship between digital finance development and corporate sustainable development.” c)what benefit have investors if they decide to use the proposed approach in their digital finance: “These findings suggest that with the rapid emerging of fintech in China, digital finance is able to provide more support for firms, especially for these non-state-owned, small and medium-sized enterprises, as well as those located in underdeveloped areas, to stabilize financial situation and improve operational capabilities in pursuit of sustainable development. ”

Q8: The methodology section is weak. Author should clearly mention the methodology and reasons for using it. which is missing.

Response: this study conducts empirical analysis using a two-way fixed effects model. Each regression model has gone through the default robustness standard error procedure. Logarithm analysis is performed on all non-ratio variables and the time, individual, and industry fixed effects are controlled (mentioned in 3.3). In addition, in order to ensure the robustness of the regression results, this paper employed robustness tests and endogenous analysis. Specifically, System-GMM Estimation was introduced to address the reverse causality relationship between digital finance and corporate sustainability performance. Firms from high-tech industries were excluded to alleviate the potential endogeneity brought by innovation effects. Digital finance, which is highly correlated with both the scale of traditional financial institutions and the levels of regional economic development, was substituted with the ratio of the balance of financial institution loans to GDP (loan/GDP) as an alternative measure of digital finance. Moreover, considering that there may be a cubic relationship between regional digital finance and corporate sustainability performance, i.e., an N-shaped or horizontally S-shaped relationship, the cubic form of digital finance was added to the baseline model (mentioned in section 4.5).

Round 3

Reviewer 4 Report

The author has addressed almost all my comments in a professional manner. The same are also implemented in the revised manuscript. As a reviewer, I am satisfied with the reply of my comments and concerns.